# Inferring on *Speleomantes* Foraging Behavior from Gut Contents Examination

**DOI:** 10.3390/ani13172782

**Published:** 2023-08-31

**Authors:** Fabio Cianferoni, Enrico Lunghi

**Affiliations:** 1Research Institute on Terrestrial Ecosystems (IRET), National Research Council of Italy (CNR), 50019 Firenze, Italy; cianferoni.fabio@gmail.com; 2Natural History Museum of the University of Florence, “La Specola”, 50125 Firenze, Italy; 3Department of Life Health and Environmental Sciences (MeSVA), Università degli Studi dell’Aquila, 67100 L’Aquila, Italy; 4Associazione Natural Oasis, 59100 Prato, Italy; 5Unione Speleologica Calenzano, 50041 Calenzano, Italy

**Keywords:** cave biology, dataset, Italy, predator, prey, salamander, trophic web

## Abstract

**Simple Summary:**

The foraging behavior of European cave salamanders (genus *Speleomantes*) is discussed through more detailed considerations starting from published datasets gathering prey recognized from their gut contents. Flying insects were consumed the most, with a minor quantity of elongated prey. The scarce occurrence of strictly-cave prey allows us to hypothesize that *Speleomantes* mainly forage in surface environments, while the presence of aquatic invertebrates in the diet suggests the hypothesis of direct predatory activity in shallow water bodies. The morphology of the prey (e.g., size or presence of long appendages) seem to be a feature influencing *Speleomantes* prey choice, while chemical or mechanical defenses of some invertebrates do not appear to be a real limit for these salamanders.

**Abstract:**

We here provide the first comprehensive analysis and discussion on prey consumed by the European cave salamanders of the genus *Speleomantes*. Our study stems from the need to shed light on the still unknown foraging behavior adopted by *Speleomantes* cave salamanders. Starting from the published datasets on gut contents from all *Speleomantes* species (including hybrids), we here discuss additional information (i.e., species ecology, lower taxonomic level), which were systematically omitted from those data sets. We analyzed a data set consisting of 17,630 records from 49 categories of consumed prey recognized from gut contents of 2060 adults and juveniles *Speleomantes*. Flying prey accounted for more than 58% of the prey items, while elongated crawling prey accounted for no more than 16% of the diet within a single population. Among the total recognized prey items, only three can be surely ascribed to the group of strictly-cave species (i.e., troglobites), meaning that European cave salamanders mostly forage in surface environment, and therefore represent one of the major drivers of allochthonous organic matter in subterranean environments. Some of the consumed prey seemed to be aquatic, allowing us to hypothesize whether *Speleomantes* are able to catch prey from a shallow body water. Furthermore, European cave salamanders possess the ability to prey upon taxa characterized by particular anti-predator defenses, while morphological constraints seem to be the most important limit to prey consumption. For each specific case, we provide insights and propose hypotheses concerning the foraging behavior that need to be tested to properly understand the foraging behavior of this cryptic salamanders.

## 1. Introduction

The European cave salamanders of the genus *Speleomantes* are the only representative of the Plethodontid family in Europe [1]. *Speleomantes* is a group of allopatric species endemic to the Italian peninsula and Sardinia, and to a small part of French Provence [1]. Three species, *S. strinatii*, *S. ambrosii*, and *S. italicus*, are distributed in mainland Italy; the former is the only species present in France, while the latter also occurs in the Republic of San Marino [1]. Five other species, *S. flavus*, *S. supramontis*, *S. imperialis*, *S. sarrabusensis,* and *S. genei*, are endemic to Sardinia island, where their distribution is mostly shaped by the geomorphologic features of the island [2]. So far, two contact zones where mainland species give birth to viable hybrids are known [3,4]. A few cases of introduction are also known. In France, besides the autochthonous populations of *S. strinatii*, there are at least two introduced populations, one in the center of the country and one in the Pyrenees [5,6]. Additionally, a population of *S. italicus* has been introduced in the north-western part of Germany [7,8]. In a few cases, mainland *Speleomantes* were also moved within Italian territory for scientific purposes [1,9]. *Speleomantes* are facultative cave species able to maintain stable populations in subterranean environments [10,11], where they avoid external unsuitable climatic conditions (too hot and dry) and lower their predation risk [12,13,14]. In surface environments, they can be usually found in forested areas or in artificial structures (such as springs, cellars, and dry stone walls), being active mostly at night and when suitable microclimatic conditions occur [15,16,17].

*Speleomantes* are generalist predators consuming a large variety of different prey [18,19]. They are able to prey under lighted condition as well as in complete darkness, using a combination of visual and chemical cues to locate prey [20]. When they approach a potential prey, *Speleomantes* “shoot” their protrusible tongue furnished with a sticky pad and hit the target in a fraction of a second [21,22]. The extreme speed of this action, combined with the cryptic behavior of *Speleomantes*, make observations of their foraging behavior difficult in the wild. The trophic niche and foraging behavior of *Speleomantes* is only known from analyses of gut contents [23,24]. Researchers have discovered significant inter- and intraspecific variability of the trophic spectrum of *Speleomantes*, identifying substantial seasonal variations of their diet [23,25,26,27]. Nonetheless, different behavioral traits also emerged from those studies, such as divergences in the number and diversity of consumed prey [25], as well as different degrees of diet specialization of individuals among species and populations [28,29]. Besides that, analyses on specific prey ecology, and thus using information from a lower taxonomic level, are still lacking, and this hampers an expansion on our knowledge on the foraging behavior of *Speleomantes*.

In the current study, we used qualitative and quantitative data on consumed prey by *Speleomantes* to infer on their foraging behavior, aiming to pave the way for further studies that can test or expand upon our hypotheses. Our ambitious and unconventional methodology opposes the mainstream experimental methods where hypotheses should be set a priori, but uses the gained experience to critically observe natural events and to further develop related hypotheses that need to be tested [30]. Our idea stems from the fact that besides the growing number of studies on the *Speleomantes* diet, none of them considered direct observations of salamanders foraging in the wild, but they only analyzed gut contents obtained from captured individuals [23,25,31,32]. Partially digested prey are hard to recognize and therefore it has been conventionally chosen to provide information up to their taxonomic order, to maximize the confidence of prey recognition and create standardized data sets [18,23]. However, in some circumstances additional information on consumed prey can be obtained (e.g., lower taxonomic level, ecology). The objective of our study was to use field observations and higher taxonomic resolution of prey omitted from previous studies of *Speleomantes* to provide additional information on the foraging behavior of these cryptic salamanders.

## 2. Materials and Methods

We analyzed published data sets collecting information on the prey items recognized from stomach contents obtained from all *Speleomantes* species [18,19,32,33,34,35]. In our discussion of results, we also include unpublished materials. *Speleomantes* were opportunistically captured inside caves and other artificial subterranean environments, or in forested areas and inside dry stone walls. Stomach contents were then collected from individuals with snout-vent length > 40 mm through stomach flushing [36]. Stomach contents were preserved in 70% ethanol until observed at microscope [18]. These data sets report the taxonomic order of the recognized prey, except for a few cases in which family or developmental stages (larva vs. adult) are also shown. Occasionally, the authors of the current study were able to collect additional taxonomic and/or ecological information on the consumed prey while building up those data sets; these further details are here considered and discussed to infer on the foraging behavior of *Speleomantes* cave salamanders. The analyzed dataset consisted of 17,630 individual prey items from 49 different categories which included ordinal identifications, larval stage, and in a couple of cases, distinctive morphology and ecology for the families Staphylinidae and Formicidae [18] (Appendix A). We excluded from the general analysis the prey categories related to *Speleomantes* eggs/skin/individuals, as these are exceptional food items individuals probably consider when particular scarcity of trophic resources occur [32,37]. However, we briefly discuss the case of potential cannibalism in a separate paragraph. The samples analyzed in the current study are stored at the Natural History Museum of the University of Florence.

To perform a further qualitative analysis on this data set, we identified three additional integrative categories defined by movement techniques of prey. Specifically, the new categories are as follows: strong flyers, taxa that mostly fly when they move (Ephemeroptera, Hymenoptera, Mecoptera, Trichoptera, Plecoptera, Lepidoptera, Diptera); occasional flyers, species that mostly crawl but can also fly (Hemiptera, Coleoptera, Coleoptera_Staphylinidae); non-flying invertebrates, all the remaining prey that only crawl on surfaces, which includes larval stages [18]. An additional category here defined as “elongate”, including all crawling taxa with body at least four times longer than its width (Lithobiomorpha, Geophilomorpha, Scolopendromorpha, Julida, Polydesmida, Pulmonata slugs, Gordioidea (Gordea), Coleoptera larva, Lepidoptera larva, Neuroptera, Diptera larva, Haplotaxida), was also considered.

## 3. Results and Discussion

### 3.1. Flying vs. Walking Prey

The first obvious deduction that can be drawn from the available datasets is that *Speleomantes* primarily consumed flying prey; strong flyers represented 58.14% of the recognized prey (Appendix A). This is quite curious as the soil probably offers a larger amount of different prey which are likely slower and therefore easier to catch than flying ones. The evolution of the hyper-fast protrusible tongue in *Speleomantes* may have been promoted by selecting this particular group of prey [38,39].

*Speleomantes* are among the plethodontid species that spend a large portion of their life climbing and clinging to vertical surfaces [40,41]. Indeed, when in subterranean environments they are commonly found on the cave walls, while in epigeous environments they often climb on rocks and trees [42,43,44]. This habitus probably gives *Speleomantes* the opportunity to avoid most of their terrestrial predators [1,45,46]. Therefore, spending less time on the ground allowed *Speleomantes* to reduce their consumption of crawling prey, and developed an affinity for taxa that mostly rest on cave walls (e.g., crane flies and other Diptera) or on the trees (e.g., bark beetles of the family Curculionidae, and flat bugs of the family Aradidae living under the bark of trees). This hypothesis is supported by the study of Roth [47], which observed an evident optical preference for squared prey moving vertically (shape similar to a fly) rather than with rectangular shape moving horizontally (“wormlike” prey) in two *Speleomantes* species. The number of elongate (wormlike) prey consumed by *Speleomantes* was generally low (6.36%) but increased in surface populations, although remaining a small fraction of the overall diet (5.16% of the recognized prey in subterranean populations vs. 16.05% in the surface populations) (Appendix A). Analyzing the stomach contents of the sympatric fire salamander (*Salamandra salamandra*) for a forest population of *S. italicus* [48], elongate prey represented 58.2% of the consumed prey for this species, allowing to confidently assume that there was not a shortage of such prey but probably *Speleomantes* consumed different ones. An experiment in which both prey typologies and abundances are experimentally controlled may help in establishing whether preference rather than opportunism is the major driver of prey selection in *Speleomantes*.

A further consideration can be made on the advantage that catching prey with a protrusible tongue may represent for *Speleomantes*. To increase protection against potential predators [49,50], *Speleomantes* usually seek refuge in holes, cracks, and cervices present on cave walls, rocks, and trees. Indeed, to increase the protection of their brood, gravid females commonly choose to lay their eggs in hidden and covered places [51,52]. Eggs need around 4–6 months to hatch and females rarely leave the nest unattended during this period, being therefore subjected to prolonged starvation [1,53]. Consequently, being able to catch passing prey from hidden places may provide substantial advantages to females that are protecting their nests or, more generally, to hidden individuals that are avoiding potential treats.

### 3.2. Affinity for Surface Species or Opportunism?

*Speleomantes* are characterized as opportunistic predators preying on a wide diversity of invertebrate species [18,19]. Most of the populations included in gut contents analysis were from subterranean environments (38 out 40). The higher number of consumed prey were crane flies (Diptera, Tipuloidea). These diptera are commonly found in most subterranean environments, particularly in the areas of the entrance and twilight zone [54,55], where they can reach very high densities [56]. These dense clusters of potential prey represent an optimal condition for *Speleomantes* to catch as many prey as possible with minimum effort [57]. Gregarious species are particularly attractive for *Speleomantes*, and there is no surprise if most individuals from subterranean populations, during the hot season, basically feed only on crane flies [18,19,23,32]. Similarly, inside artificial springs where a population of *S. sarrabusensis* occurs, small rove beetles (Coleoptera Staphylinidae) usually form dense groups; indeed, rove beetles are the most commonly consumed prey by this population [25]. Both crane flies and rove beetles recognized from *Speleomantes* stomach contents were mainly epigean taxa that moved underground to avoid the harsher climatic summer conditions [12]. Other than a few exceptions of confirmed prey troglomorphism (*sensu* Christiansen [58]), all the other prey consumed by *Speleomantes* were epigean species. *Speleomantes* are facultative cave species able to forage both inside and outside the cave [59]. Surface environments are characterized by a greater amount of food resources compared to subterranean ones, where the peculiar environmental conditions drastically reduce the diversity and the abundance of inhabiting species [60]. Therefore, *Speleomantes* probably prefer to forage in surface environments (or nearby the cave entrance) where the food supply is the highest. Over 12,587 prey items consumed by subterranean populations, we were able to identify only three types of prey with troglomorphic characters: a depigmented planaria of the genus *Dendrocoelum*, a cave beetle of the genus *Duvalius* (Coleoptera Carabidae), and a blind beetle of the family Curculionidae. The negligible presence of cave adapted species within the prey consumed by *Speleomantes* allows to hypothesize that these salamanders do not exert substantial predatory pressure on subterranean species (to the benefit of the numerically rare cave-adapted species), but they are rather important drivers of allochthonous organic matter useful to sustain the entire subterranean ecosystem [54,61]. The use of stable isotopes and metabarcoding on *Speleomantes* stomach contents may help in providing a clearer overview of the typologies of prey consumed by these salamanders. Furthermore, quantitative and qualitative analyses on *Speleomantes* excrements in caves would be helpful to better comprehend the salamanders’ contribution of allochthonous materials from the surface to the subterranean community.

In surface populations of *Speleomantes*, the consumption of springtails (Collembola) was higher compared to subterranean ones (23.6% in forests vs. 11.3% in caves). Collembola are a widespread species in both surface and subterranean environments [62,63], although no comparative information on their abundances exist. All taxa preyed by *Speleomantes* are likely more abundant in surface environments rather than subterranean ones, although we do not know the magnitude of such increase for single groups. Interestingly, springtails are among the smallest prey consumed by *Speleomantes* [25] and they do not show particular gregarious behavior, meaning that every consumed Collembola was individually captured. In some *Speleomantes* from surface environments, we were able to recognize at least 3 or 4 different morphospecies belonging to the order Symphypleona, with additional information supporting the hypothesis of the specific intention of salamanders to prey upon springtails. Catching each single springtail might be convenient for *Speleomantes*, if not for the positive balance of energy (energy used to prey research and for tongue activation vs. energy gained) at least in terms of intake of important elements. Studies on *Speleomantes* metabolism, as well as on the nutritional intake provided by the different prey typologies, may be useful to better comprehend the prey selection performed by individuals [28].

### 3.3. Do Speleomantes Catch Prey from Aquatic Environments?

Analyzing the stomach contents of a fully terrestrial salamander, we would not expect to observe aquatic prey taxa. However, a few sporadic exceptions exist. Residuals of a Hemiptera belonging to the family Veliidae have been found in an individuals of *S. strinatii* from the Pyrenees (France) [33]. This population inhabits a mine with an inner waterbody, from where the prey could probably have been caught. However, these species can walk on the surface of water and can also be found on emergent vegetation or on the banks, acting as terrestrial-like taxa. From the stomach contents of an individual of *S. ambrosii*, a flatworm of the genus *Dendrocoelum* (Platyhelminthes Tricladida) has been recognized [26]. Flatworms are aquatic species that crawl on the bottom of bodies of waters. This planarian inhabits a few ponds inside a cave located in Liguria (north-west Italy) [64]. Adult diving beetles (Coleoptera Dytiscidae) have been collected from the stomach contents of individuals of *S. italicus*. This population of *S. italicus* occurs in a sinkhole located at the top of the mountain, where water bodies are basically absent; there are only a few small ponds made of dripping water inside the cave. Helophoridae beetles (adult individuals) have been recognized from *S. italicus* stomach contents; some species belonging to this family are also aquatic. A similar case is provided by the larva of Trichoptera, another aquatic prey recognized from the stomach contents of six *Speleomantes* populations (four of *S. italicus*, one each of *S. genei* and *S. supramontis*), of which only one is from surface environments. In these instances of aquatic prey, salamanders likely hunted along the edge of shallow waters, but it is not clear whether the prey were consumed along the edge or if the salamander entered the water. *Speleomantes* are in fact able to easily swim towards the bank when they fall into a shallow body water (Lunghi pers. obs.). It cannot be totally excluded, at least in the case of some of these types of prey, that the salamanders encountered them during a short “terrestrial phase”. In fact, it is possible that flatworms can also exploit the water film on cave walls, for example if they are flowing inside the cave from the dendritic fissures filled of water of the epikarst. Adult diving beetles are also able to fly towards a new pond if necessary [65]; therefore, they can become a temporary terrestrial target. Even adult helophorids occur mainly in the peripheral parts of bodies of water, even in the mud on the banks [66]. Nonetheless, these taxa need frequent surfacing to store atmospheric air for respiration. Therefore, most of the aquatic taxa found within the stomach contents of *Speleomantes* are taxa able to exploit (even for a short time) terrestrial environments or that can occur in very shallow water, often near the shores. However, considering that these prey have been recognized from multiple individuals, it seems unlikely that they have always been intercepted by salamanders during occasional movements out of the water. This is also corroborated by the discovery of more than one immature specimen of caddisfly. In fact, the larvae of Trichoptera have gills in this phase and do not leave the aquatic environment [67]. Although we have no direct evidence for a potential foraging in water, this hypothesis appears quite robust through such evidence as salamanders foraging in water, and deserves consideration in future studies.

### 3.4. “Unfriendly” Prey

Among the consumed prey by *Speleomantes*, some are characterized by chemical defenses, mechanical defenses, or a combination of both. Ants (Hymenoptera, Formicidae) (7.07%) can either bite or secrete formic acid in defense [68,69]; millipedes (Julida and Polydesmida) (1.80%) can curl up as a defense position and secrete irritating chemicals to deter predators [70]; moths (Lepidoptera) (0.16%) can produce specific alkaloids which can even deter spiders to prey on them [71,72] (Appendix A). Therefore, it seems that *Speleomantes* may have some resistance against these kinds of defenses, especially for ants which represented a relatively highly consumed prey. The large number of consumed ants may be due to the tendency of high sociality for this taxon, meaning that ants are seldom found singularly in the environment. Targeted studies are certainly necessary to deepen the matter further.

Overall, only 120 Pulmonata have been recognized from the analyzed data sets (0.68%) (Appendix A). Interestingly, only six were slugs, recognized from four individuals in an epigeous population of *S. italicus* [48]. The remaining prey of the category were small snails (with external shell). These cave salamanders are able to swallow and handle small-sized land snails; their shell often remains intact during the digestive process and it can be easily recognized from gut contents. On the other hand, the low number of consumed slugs may be due to their highly viscous mucus which is often used as anti-predatory defense [73]; in this specific case, the mucus may hamper an easy swallowing of the prey by *Speleomantes*. This hypothesis needs to be tested to rule out the possibility that faster digestion of soft body prey masked the presence of these taxa within *Speleomantes* stomach contents.

When underground, *Speleomantes* are top predators of the local food web [74]; however, some arthropods (Scolopendromorpha, Lithobiomorpha, Araneae) can also represent potential predators for *Speleomantes*, especially for juveniles. This is the case, for example, for the large spiders of the genus *Tegenaria* (Agelenidae) and *Meta* (Tetragnathidae), which can trap and forage on juvenile *Speleomantes* in to their webs [49,50]. Among the 684 (3.88%) records of consumed Araneae (Appendix A), none of the specimens could be ascribed to the families Agelenidae or Tetragnathidae. Although quite rare, centipedes (Scolopendromorpha and Lithobiomorpha) are also able to actively prey on juvenile *Speleomantes*. One case has been documented by Sanna et al. [75], where *Plutonium zwierleini* (Scolopendromorpha) was observed holding and transporting a juvenile of *S. supramontis*.

Orthoptera were consumed only in 41 cases (0.23%) (Appendix A), in which specimens were mostly ascribed to the genus *Gryllomorpha*, dorsoventrally flattened crickets of relatively small size (<20 mm) [76]. A single individual belonging to the cave cricket *Dolichopoda laetitiae* was recognized. The long and robust appendages of these crickets likely represent a physical barrier that prevent them from being ingested by *Speleomantes*. This is particularly true for *D. laetitiae*, which has extremely elongated appendages, an evident adaptation to subterranean life [77]. *Dolichopoda* cave crickets often occur with high abundances in caves, especially during periods in which nymphs are present [78,79]. Interestingly, although being of a size of few millimeters, neither *Dolichopoda* nymphs seem to represent suitable prey for *Speleomantes*.

### 3.5. The Myth of Cannibalism

Besides the controversial observation of cannibalism reported in the review of Lanza [1], only in a single individual (a female of *S. ambrosii*), among the 2060 investigated, has been found with residuals of a juvenile in its gut contents [18]. The authors proposed that the consumed juvenile was already dead (and probably already partially decomposed) [25] and the female just consumed the remnants of its body. Indeed, *Speleomantes* usually recycle their own organic matter, like unfertilized eggs or skin after molt [1,25]. This tendency of recycling all possible organic matter is advantageous for populations inhabiting subterranean environments, were food resources are scarce [54]. The odd observation was reported in Lanza as «The «mysterious» disappearance of some very small Speleomantes reared… without feeding, together with adults in small boxes at 4–5 °C» [1] (p. 47) can be therefore considered a random event due to unnatural artificial conditions, similarly to what has been thought about the reproductive modality adopted by *S. sarrabusensis* [80].

### 3.6. Do Speleomantes Process Their Food?

In a recent study, Spence et al. [81] showed that the Axolotl (*Ambystoma mexicanum*) is able to process its food through intraoral behaviors, referring to as “chewing-like” behavior. The name of the family Plethodontidae origins from Ancient Greek and refers to the large number of teeth observed in these salamanders (plêthos, “great number” + odoús, “tooth”) [1]. However, as far as we know, these species do not seem to adopt such behaviors. As we stated, *Speleomantes* capture prey using their protrusible tongue [20,21], and swallows them entirely. Indeed, among the thousands of recognized examples of prey, we never found any sign of mechanical food processing, but we rather often found whole prey. This allows to hypothesize that *Speleomantes* do not mechanically process their food, but they rather swallow and only digest using gastric juices. Teeth could only have the function of holding the prey. However, at the moment this remains a plausible explanation which lacks support from empirical studies.

An interesting case is represented by a Horsehair worm (Gordioidea) that was found alive among the stomach contents of an *S. ambrosii* individual. Horsehair worms parasite other invertebrates, but they gather in water bodies for reproduction. It is possible that this parasite was inside a host (e.g., a coleoptera found within the same stomach contents) from which it exited once the prey was already ingested. The alternative is that the free-living horsehair worm was directly caught from a shallow water body (see above) and swallowed alive. Independently from how the prey entered the salamander stomach, the resulting related scenario is a series of interesting hypotheses that should be addressed. Do *Speleomantes* lack mechanical process of their food? How long prey remain alive in their stomach? How does their digestion work?

## 4. Conclusions

In the current study, we provide a deductive discussion on the potential foraging behavior of *Speleomantes*. The more fine-scale examination of *Speleomantes* gut contents adopted here allowed us to infer specific behaviors of these salamanders and, accordingly, to develop interesting hypotheses. Besides presenting evidence contrary to controversial claims, we suggested potential experiments that would help in better comprehending many aspects that are still obscure regarding the foraging behavior of *Speleomantes*.

## Data Availability

We here qualitatively discuss data sets already published. The related references are all cited in the text.

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
