# Peer review of "Inferring on *Speleomantes* Foraging Behavior from Gut Contents Examination"

_animals, 2023, doi:10.3390/ani13172782_

Round 1

Reviewer 1 Report

The term "preference" is misused when identifying the primary prey consumed by salamanders. The availability of prey would need to be quantified and compared with prey use (i.e., consumption) in order to test for prey preferences (as was suggested in lines 145-147 in the manuscript). See suggestions in the attached review.

The term "species" is frequently used in the manuscript but prey items were identified at the ordinal or familial levels. As such, the term "taxa" should be used instead.

A table showing the cumulative numbers from gut content analyses and the corresponding percentages for prey taxa and categories would be useful to the reader.

Some suggestions for sentence structure and word choice are provided in the attached review.

Author Response

Thank you for positive comments and for the effort in suggesting all changes in spelling

Reviewer 2 Report

1. This study provide a results of deductive discussion on the potential foraging behavior of Speleomantes.  Analysis of a dataset gathering 17,627 records 27 from 49 categories of consumed prey recognized from gut contents of 2,060 adults and juveniles Speleomantes allowed the authors to analyze specific behaviors of these salamanders and, accordingly, to develop interesting hypotheses.  They suggested potential experiments that would help in better comprehending many aspects that are still obscure of Speleomantes foraging behavior. In this study we provide a deductive discussion on the potential foraging behavior of Speleomantes.  

2. The hypothesis is interesting and looks correct.  I would like to see more detailed knowledges about used qualitative and quantitative data a presented as the tables and figures. 

3. The absence of the tables and figures presented the data collected including the own data and  clarifying the content of the paper. 

Author Response

Thanks for positive comments
